# A combination of oxygenation and driving pressure can provide valuable information in predicting the risk of mortality in ARDS patients

Yu-Yi Yu[1,2], Wei-Fan Ou[3], Jia-Jun Wu[3,4], Han-Shui Hsu[1,5], Chieh-Laing Wu[2,6], Kuang-Yao Yang[1,7,8], Ming-Cheng Chan[2,3,6]*

1 Institute of Emergency and Critical Care Medicine, National Yang Ming Chiao Tung University, Taipei, Taiwan, 2 Department of Critical Care Medicine, Taichung Veterans General Hospital, Taichung, Taiwan, 3 Division of Chest Medicine, Department of Internal Medicine, Taichung Veterans General Hospital, Taichung, Taiwan, 4 Division of Pulmonary Medicine, Department of Internal Medicine, Chung Shan Medical University Hospital, Taichung, Taiwan, 5 Department of Thoracic Surgery, Taipei Veterans General Hospital, Taipei, Taiwan, 6 School of Post BaccalaureateMedicine, National Chung Hsing University, Taichung, Taiwan, 7 Department of Chest Medicine, Taipei Veterans General Hospital, Taipei, Taiwan, 8 School of Medicine, National Yang Ming Chiao Tung University, Taipei, Taiwan

* mingcheng.chan@gmail.com

**Data Availability Statement:** All relevant data are within the manuscript and its Supporting information files.

## Abstract

### Background

Acute respiratory distress syndrome (ARDS) is a common life-threatening condition in critically ill patients. Itis also an important public health issue because it can cause substantial mortality and health care burden worldwide. The objective of this study was to investigate therisk *factors that impact ARDS mortality* in a medical center in Taiwan.

### Methods

This was a single center, observational study thatretrospectively analyzed data from adults in 6 intensive care units (ICUs) at Taichung Veterans General Hospital in Taiwan from 1st October, 2018to30th September, 2019. Patients needing invasive mechanical ventilation and meeting the Berlin definition criteria were included for analysis.

### Results

A total of 1,778 subjects were screened in 6 adult ICUs and 370 patients fulfilled the criteria of ARDS in the first 24 hours of the ICU admission. Among these patients, the prevalenceof ARDS was 20.8% and the overall hospital mortality rate was 42.2%. The mortality rates of mild, moderate and severe ARDS were 35.9%, 43.9% and 46.5%, respectively. In a multivariate logistic regression model, combination of driving pressure (DP) > 14cmH$_2$O and oxygenation (P/F ratio)≤150 was an independent predictor of mortality (OR2.497, 95% CI 1.201–5.191, $p$ = 0.014). Patients with worse oxygenation and a higher driving pressure had the highest hospital mortality rate($p$<0.0001).

**Funding:** This study was supported by Taichung Veterans General Hospital (TCVGH-1104101C). The funder provided the grant for project administration, material and supplies costs, and assistant salary. The WFO, JJW, CLW, and MCC are employees of TCVGH. The funder provided partial salary for the author YYY. The funder did not have any additional role in the study design, data collection and analysis, the decision to publish, or the preparation of the manuscript.

**Competing interests:** The authors have declared that no competing interests exist.

**Abbreviations:** ABG, Artery blood gas; APCHE II, Acute Physiology and Chronic Health Evaluation II; *ARDS, Acute respiratory distress syndrome*; BMI, *Body Mass Index*; CCI, *Charlson Comorbidity Index*; CI, Confidence interval; DP, Driving pressure; FiO₂, Inspired fraction of oxygen; *ICU, Intensive care unit*; IQR, Interquartile range; MAP, Mean Airway Pressure; NMBAs, Neuromuscular blocking agents; OR, Adjusted odds ratio; P/F ratio, Ratio of partial pressure of oxygen to fraction of inspired oxygen; PaO₂, Partial pressure of oxygen; PEEP, Positive end-expiratory pressure; PIP, Peak inspiratory pressure; P$_{plat}$, Plateau pressure; RR, Respiratory rate; SD, Standard deviation; SOFA, The sequential organ failure assessment score; TRALI, Transfusion-related acute lung injury; VT, Tidal volume.

## Conclusions

ARDS is common in ICUs and the mortality rate remains high. Combining oxygenation and respiratory mechanics may better predict the outcomes of these ARDS patients.

## Introduction

Acute respiratory distress syndrome (ARDS) is a life-threatening condition which may result from a variety of pulmonary (e.g., pneumonia and aspiration) and extra-pulmonary causes (e.g., sepsis, trauma and pancreatitis) [1, 2]. ARDS is characterized by acute, diffuse alveolar inflammation and flooding, leading to increased capillary permeability, which results in lung tissue edema and loss of aeration [2–4]. Clinically, patients with ARDS often present with hypoxemia, pulmonary congestion, and decreased respiratory compliance [1, 5]. Patients with ARDS often need invasive mechanical ventilation to rescue their hypoxemia. Although there have been recent advances in the management of ARDS, including protective mechanical ventilation [6], prone positioning [7] and extracorporeal life support [8], the mortality rate remains high [9]. In addition, patients who survive ARDS may also suffer from significant sustained disabilities, both physically and psychologically, leading to a decreased quality of life and even increased mortality in later years [10, 11].

ARDS is common in critically ill patients who are admitted to intensive care units (ICUs), but often goes unrecognized and undertreated [9]. In the United States, ARDS affects approximately 200,000 patients per year and results in about 75,000 deaths annually [12]. ARDS is also an important public health issue as it is a common cause of death for severe respiratory system infections, especially influenza [13] and coronavirus disease 2019 (COVID-19) [14, 15].

In Taiwan, the epidemiological data, patterns of care and outcomes of patients with ARDS are not clear. Given that ARDS has an important impact on public health burden, we conducted a retrospective observational study to investigate its prevalenceand factors associated with the outcomes of critically ill patients with ARDS.

## Method

### Patient and study design

This was an observational study of six adult ICUs in a single tertiary referral institute in central Taiwan. The ICUs were functionally divided into medical, respiratory, neurological, surgical, cardiovascular and trauma units. We retrospectively analyzed the data of critically ill patients with ARDS who were admitted to the adult ICUs of Taichung Veterans General Hospital (TCVGH) from the 1$^{st}$ October, 2018 to the 30$^{th}$ September, 2019. Adult patients (age >20 years old) who were admitted to the ICUs needing invasive mechanical ventilation and who met the Berlin definition criteria for ARDS [16] were included for analysis. Patients with ARDS were identified by a respiratory therapist and verified by two critical care physicians with a respiratory specialty. During the study period, we retrospectively collected data from a quality improvement program via a clinical audit of daily practice in patients with acute respiratory failure who needed mechanical ventilation in the ICUs. This study was approved by the Institutional Review Board of Taichung Veterans General Hospital Taiwan (IRB number: CE20049B). Data were collected and accessed withthe IRB approval on 21$^{st}$ February, 2020. Given that the research was limited to the secondary use of data previously collected during daily practice and the fact that the patients' identities were completely anonymized, the requirement for informed consent from the study subjects was waived by the IRB of Taichung

Veterans General Hospital Taiwan due to no more than minimal risk to the subjects. All methods were carried out in accordance with the IRB's guidelines and regulations.

## Types of outcome measures

Our primary outcome was to investigate factors that impact ARDS mortality. The secondary outcome was to understand the real-life practice pattern in management ofARDS in Taiwan.

## Data collection

All data were collected from medical records, including the patients' demographic data, co-morbidities, and basic laboratory tests. Disease severity scores, including Acute Physiology and Chronic Health Evaluation II (APACHE II) [17], and Sequential Organ Failure Assessment (SOFA) scores [18] were determined on the first, third and seventh days of the ICU admission. Mechanical ventilation parameters were also collected, including fraction of inspired oxygen ($FiO_2$), positive end-expiratory pressure (PEEP, $cmH_2O$), tidal volume (VT) adjusted by predicted body weight (PBW), plateau pressure ($P_{plat}$) and peak airway pressure ($P_{peak}$) on the first three days of the ICU admission. $P_{plat}$ was measured at end-inspiration by a 0.5 to 1 second inspiration hold on a mechanical ventilator within 24 hours of the initiation of mechanical ventilation. Driving pressure was calculated as the difference between $P_{plat}$ and PEEP [19].

Information on major co-morbidities, including cardiovascular disease, cerebrovascular disease, dementia, chronic pulmonary disease, rheumatic disease and malignancy was collected. The Charlsoncomorbidity index (CCI) [20] was calculated using the International Classification of Disease 10[th] Revision [21] diagnosis codes from the patients' medical records. Diabetes was defined by medical history and a laboratory examination with HbA1c >6.5%. The severity of ARDS was classified by the$PaO_2$/$FiO_2$ (P/F)ratio according to the Berlin definition [16]. Mechanical ventilator parameters, including mode, $FiO_2$, VT, PEEP, respiratory rate (RR), peak inspiratory pressure (PIP), and $P_{plat}$ were collected in the first 24 hours of the ICU admission. ARDS severity was determined using the worst partial pressure of oxygen to fraction of inspired oxygen ratio within the first 24 hours following ARDS diagnosis.

## Statistical analysis

Data is presented as frequency (percentages) for categorical variables and as mean ± standard deviation or median (interquartile range [IQR]) for continuous variables. Differences between alive and dead groups were determined using the Student's t-test and the Mann-Whitney U test for continuous variables and the Chi-squared test for categorical variables. One-way analysis of variance was used to compare variables among the three ARDS severity groups and the-Pearson's Chi Square test to compare categorical variables among the three groups of ARDS severity. Kaplan-Meier analysis was performed to test the mortality of driving pressure above or below 14 $cmH_2O$ in different ARDS severities. The log-rank test was used to compare mortality between two groups. A multivariate logistic regression model was conducted to identify independent variables that predicted mortality. Statistical significance was set at a two-sided p value of <0.05. All data were analyzed using SPSS software version 22.0 (SPSS Inc., IBM Corp, Armonk, NY, USA).

## Results

### Patient enrollment and clinical features

Of 1,778 patients with acute respiratory failure who needed invasive mechanical ventilation, 370 (20.8%) fulfilled the criteria of ARDS in the first 24 hours ofthe ICU admission (Fig 1).

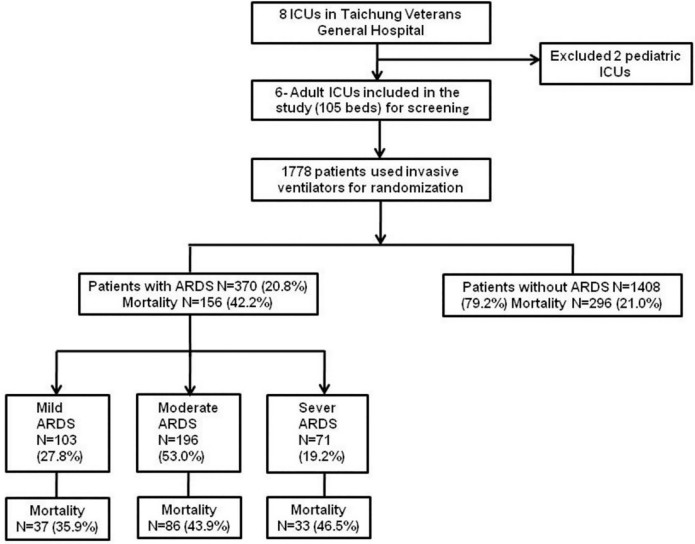

**Fig 1. Flow chart of patient enrollment.**

The hospital mortality of patients with ARDS was 42.2%, while in those without ARDS, it was just 21.0%. In this cohort study of ARDS patients, the average age was 67.7 years old. There were more male (68.1%) than female (31.9%) patients. Most of these ARDS patients were admitted via the emergency room (81.9%) and there were more in the medical (79.7%) ICU compared with the surgical (20.3%) ICU. The overall disease severity in this cohort study of ARDS patients was high in terms of APACHE II and SOFA scores at admission and on the following ICU days. Co-morbidities were common among ARDS patients. Diabetes, renal disease and malignancy were the most common co-morbidities in this cohort study. The etiology of ARDS came primarily from pulmonary (80.5%) causes as opposed to extra-pulmonary (19.5%) causes. The leading causes of ARDS were pneumonia and aspiration for pulmonary ARDS and sepsis for extra-pulmonary ARDS. ADRS patients also consumed a substantial amount of medical resources, with long ICU stays (13, 8–22 days), hospital stays (29.5, 18–46 days) and ventilator days (13, 7–24 days) (Table 1).

## ARDS severity

In this cohort study of 370 patients, 103 (27.8%) had mild ARDS, 196 (53%) had moderate and 71 had severe (19.2%). The increasing severity of ARDS was paralleled by a worsening of the APCHE II and SOFA scores at admission and on the following ICU days. There wereno significant difference in age, sex, admission source, ICU type, CCI and etiology among the three severity groups. Patients with higherARDS severity had a higher body mass index (BMI). Among the co-morbidities, dementia was more common in milder ARDS patients. The clinical outcomes, including ICU stay, hospital stay, ventilatordays and hospital mortality, were not significantly different among the three groups (S1 Table).

## Mechanical ventilation and adjunctive treatments in ARDS

Mechanical ventilation management on the first day of the ICU admission varies with ARDS severity. Volume-targeted ventilation (82.4%) was preferred in this cohort study of patients. Regarding the ventilator settings, FiO$_2$, PEEP and respiratory rate increased with ARDS

**Table 1. Demographic data of ARDS patients categorized by hospital mortality.**

| Characteristics | All | Survivor | Non-survivor | PValue [ab] |
|---|---|---|---|---|
| | N = 370 | N = 214 (57.8%) | N = 156 (42.2%) | |
| Age (years) | 67.7 ± 16.9 | 68.4 ± 16.6 | 66.8 ± 17.3 | 0.313 |
| Male, No. (%) | 252 (68.1%) | 148 (69.16%) | 108 (69.23%) | 0.611 |
| BMI (kg/m$^2$) | 23.7 ± 4.9 | 24.1 ± 4.9 | 23.2 ± 9.0 | 0.102 |
| Admission source, No. (%) | | | | |
| Emergency Room (ER) | 303 (81.9%) | 177 (82.7%) | 126 (80.8%) | 0.461 |
| Non-Emergency Room | 67(18.1%) | 37 (17.3%) | 30 (19.2%) | |
| Type of ICU, No. (%) | | | | |
| Medical | 295 (79.7%) | 169 (79.0%) | 126 (80.8%) | 0.67 |
| Surgical | 75 (20.3%) | 45 (21.0%) | 30 (19.2%) | |
| Severity scores | | | | |
| APACHE ll score | 28.0 ± 6.9 | 26.3 ± 6.1 | 30.4 ± 7.2 | <0.0001 |
| SOFA score, Day 1 | 10.3 ± 3.5 | 9.5 ± 3.3 | 11.4 ± 3.6 | <0.0001 |
| SOFA score, Day 3 | 9.3 ± 4.2 | 8.1 ± 3.4 | 11.2 ± 4.5 | <0.0001 |
| SOFA score, Day 7 | 7.7 ± 4.2 | 6.1 ± 3.1 | 10.4 ± 4.5 | <0.0001 |
| Comorbidities, No. (%) | | | | |
| Cardiovascular disease | 102 (27.6%) | 63 (29.4%) | 39 (25.0%) | 0.345 |
| Cerebrovascular disease | 75 (20.3%) | 48 (22.4%) | 27 (17.3%) | 0.226 |
| Dementia | 26 (7.0%) | 17 (7.9%) | 9 (5.8%) | 0.419 |
| Chronic pulmonary disease | 76 (20.5%) | 51 (23.8%) | 25 (16.0%) | 0.066 |
| Rheumatic disease | 30 (8.1%) | 11 (5.1%) | 19 (12.2%) | 0.014 |
| Hepatic disease | 67 (18.1%) | 28 (13.1%) | 39 (25.0%) | 0.003 |
| Diabetes mellitus | 145 (39.2%) | 93 (43.5%) | 52 (33.3%) | 0.049 |
| Renal disease | 127 (34.3%) | 77 (36.0%) | 50 (32.1%) | 0.432 |
| Malignancy | 141 (38.1%) | 56 (26.2%) | 85 (54.5%) | <0.0001 |
| *Charlson Comorbidity Index* (CCI) | 3.6 ± 2.9 | 3.4 ± 2.7 | 4.1 ± 3.0 | 0.006 |
| Etiology of ARDS, No. (%) | | | | |
| Pulmonary, No. (%) | 298 (80.5%) | 173 (80.8%) | 125 (80.1%) | |
| Pneumonia | 255 (68.9%) | 153 (71.5%) | 102 (65.4%) | 0.210 |
| Aspiration | 37 (10.0%) | 16 (7.5%) | 21 (13.5%) | 0.058 |
| Pulmonary contusion | 6 (1.6%) | 4 (1.9%) | 2 (1.3%) | 0.659 |
| Extrapulmonary, No. (%) | 72 (19.5%) | 50 (23.4%) | 38 (24.4%) | |
| Sepsis (non-pulmonary source) | 48 (13%) | 25 (11.7%) | 23 (14.7%) | 0.387 |
| Trauma or hemorrhagic shock | 4 (1.1%) | 4 (1.9%) | 0 (0%) | 0.086 |
| Pancreatitis | 4 (1.1%) | 1 (0.5%) | 3 (1.9%) | 0.181 |
| TRALI | 10 (2.7%) | 7 (3.3%) | 3 (1.9%) | 0.430 |
| Clinical outcomes | | | | |
| ICU length of stay | 13 (8–22) | 15 (9–22) | 11 (5–21) | 0.002* |
| Hospital length of stay | 29.5 (18–46) | 34 (24–53) | 22 (10–37) | <0.0001* |
| Ventilator-day | 13 (7–24) | 13 (7–25) | 13 (6–23) | 0.199* |

Abbreviations: *ARDS, acute respiratory distress syndrome*; BMI, *Body Mass Index; ICU, intensive care unit;* APACHE II,Acute Physiology And Chronic Health Evaluation II; SOFA, The sequential organ failure assessment score; *CCI, Charlson comorbidity index;* TRALI, transfusion-related acute lung injury;sd, standard deviation;IQR (interquartile range).

[a] *P* value represents comparisons between the survivors and non-survivors ARDS patients.

[b]*P* values for the comparison of continuous variables were calculated using both the t-test and Mann-Whitney U test.

*P values for the comparison of continuous variables were calculated using the Mann-Whitney U test

severity, whereas VT decreased. The measured pressures, including PIP and $P_{plat}$ changed in parallel with ARDS severity. The average VT was 7.5 ± 1.6 mL/kg PBW and 65.5% of these patients had ≤8mL/kg PBW. $P_{plat}$ was measured in 84.3% of patients and the average $P_{plat}$ was 21.5 ± 5.2 cm $H_2O$; 96.2% of these patients had ≤30 cm $H_2O$. A total of 64.4% of these patients received protective mechanical ventilation as defined by a VT of ≤8 mL/kg PBW and a $P_{plat}$ ≤30 cmH$_2$O. The average driving pressure was 12.9 ± 4.1 cmH$_2$O and66.4% of these patients had ≤14 cm $H_2O$. Positive end-expiratory pressure was relatively low with the average PEEP 8.8 ± 4.1 cmH$_2$O in all patients and 10.2 ± 3.0 cmH$_2$O in severe patients. We further divided the patients into four groups by DP and oxygenation. Patients with DP>14 and P/F ratio≤150 had the highest mortality rate (31/52 = 60.8%) (Table 2).

The use of adjunctive treatments in patients with ARDS was relatively low but increased with ARDS severity. In patients with severe ARDS, the percentages of patients using extracorporeal membrane oxygenation,prone position, andrecruitment maneuver were5.6%,15.5%, and 26.8%, respectively. Neuromuscular blockade was used in 43.0% of all patients and in 64.8% of severe ARDS patients (S2 Table).

## Characteristics of ARDS patients according to driving pressure

We further analyzed the characteristics of patients by driving pressure. Driving pressure at admission was available in 318 (85.9%) of the patients. Those with a driving pressure >14 cmH$_2$O had a significantly lower survival rate (33.6% vs. 52.3%, p = 0.001) (S3 Table). There were no significant difference in age, gender, BMI, co-morbidities, admission source, etiology, length of ICU stay, length of hospital stay or ventilator days between groups of driving pressure(DP)≤14cmH$_2$O and DP >14 cmH$_2$O. In ARDS patients with different DPs, the ventilator parameters including PIP and $P_{plat}$,were significantly lower in the DP ≤14cmH$_2$O group compared with the DP >14 cmH$_2$O group (p<0.0001) (S4 Table).

## Mortality of ARDS

The hospital mortality in this ARDS cohort study was 42.2% and the hospital mortality rate was paralleled by ARDS severity, which was 35.9% in mild, 43.9% in moderate and 46.5% in severe ARDS. There were no significant difference in age, sex, BMI, admission source, and type of ICU between the survivors and the non-survivors. The non-survivors had higher APACHE II and SOFA scores on the days following the ICU admission. The non-survivors also had higher CCI scores than the survivors. Among the co-morbidities, the incidence of rheumatic disease, hepatic disease and malignancy were higher in the non-survivors, but diabetes was more common in the survivors (Table 1). Among mechanical ventilator parameters, VT and PEEP were not significantly different between the survivors and the non-survivors. However, PIP, $P_{plat}$ and driving pressure were higher in the non-survivors compared with the survivors (Table 2). A multivariate regression model was adopted to adjusting for possible confounders, including prone position ventilation, aspiration of gastric content, CCI, RR, FiO$_2$, mean arterial pressure, BMI, APACHE II score andDP and P/F ratio combination groups. Andit was used to evaluate independent risk factors for hospital mortality. Two independent factors were identified, including APACHE II score (adjusted odds ratio (OR) 1.089, 95% confidence interval (CI) 1.045–1.135; $p<0.0001$) and combination of DP/oxygenation (DP>14/P/Fratio≤150, OR2.497, 95% CI 1.201–5.191, $p$ = 0.014) (Table 3).

As oxygenation (P/F ratio) is the major clinical presentation of ARDS, the severity of ARDS is traditionally classified by the P/F ratio and its impact on outcomes. Additionally, our study identified combination of oxygen and respiratory mechanics, specifically in DP>14 andP/F ratio≤150, as an independent risk factor (Table 3, p = 0.014). Compared to the group with a

**Table 2. Mechanical ventilation and arterial blood gas on admission.**

| Variables | All | Survivors | Non-survivors | PValue [ab] |
|---|---|---|---|---|
| | (N = 370) | (N = 214) | (N = 156) | |
| **Ventilator settings, first day of ARDS(Day0)** | | | | |
| Mode, No. (%) | | | | |
| Volume -targeted,No. (%) | 305(82.4%) | 181(84.6%) | 124(79.5%) | 0.257 |
| Pressure-targeted,No. (%) | 65(17.6%) | 33(15.4%) | 32(20.5%) | |
| $FiO_2$ | 0.6±0.2 | 0.63±0.22 | 0.7±0.2 | 0.022 |
| $PaO_2$(mmHg) | 103.9±53.3 | 102.7±52.8 | 105.6±54.1 | 0.609 |
| P/F ratio | 173.5±88.4 | 178.6±92.1 | 166.5±82.8 | 0.195 |
| PEEP($cmH_2o$) | 8.8±4.1 | 8.5±3.2 | 9.2±5.1 | 0.117 |
| Tidal Volume(vt/cc/kg) | 7.5±1.6 | 7.6±1.5 | 7.3±1.7 | 0.198 |
| RR(1breath/min) | 22.2±5.4 | 21.5±5.2 | 23.1±5.5 | 0.004 |
| PIP(mmHg) | 24.9±5.9 | 24.3±5.0 | 25.7±5.7 | 0.013 |
| $P_{plat}$($cmH_2o$) | 21.5±5.2 | 20.6±4.4 | 22.2±6.3 | 0.012 |
| Driving pressure($cmH_2o$) | 12.9 ± 4.1 | 12.3±3.5 | 13.8±4.8 | 0.003 |
| Compliance ($cmH_2o$) | 32.9±12.3 | 34.2 ± 11.7 | 31.0±12.9 | 0.023 |
| **Artery blood gas(ABG)-patient in ICU 24hrs** | | | | |
| pH | 7.4±0.1 | 7.38±0.89 | 7.33±0.12 | 0.319 |
| $PaO_2$(mmHg) | 122.2±98.0 | 124.6±116.2 | 119.0±66.0 | 0.589 |
| $PaCO_2$(mmHg) | 41.8±28.2 | 42.5±35.0 | 40.9±14.3 | 0.575 |
| $HCO_3$ | 21.9±5.0 | 23.6±17.0 | 22.6±20.1 | 0.587 |
| **Combination of oxygen and driving pressure** | | | | |
| DP≤14 and P/F ratio>150 | 123 (33.2%) | 82 (38.3%) | 41 (26.3%) | 0.006 |
| DP≤14 and P/F ratio≤150 | 92 (24.9%) | 58 (27.1%) | 34 (21.8%) | |
| DP>14 and P/F ratio>150 | 41 (11.1%) | 22 (10.3%) | 19 (12.2%) | |
| DP>14 and P/F ratio≤150 | 51 (13.8%) | 20 (9.3%) | 31 (19.9%) | |
| **Adjunctive Therapy, No. (%)** | | | | |
| ECMO | 7(1.9%) | 2(0.9%) | 5(3.2%) | 0.113 |
| Lung recruitment maneuver | 48(13.0%) | 23(10.7%) | 25(16.0%) | 0.136 |
| Prone position ventilation | 23(6.2%) | 9(4.2%) | 14(9.0%) | 0.061 |
| Neuromuscular blockade | 159 (43.0%) | 75 (35.0%) | 84 (53.8%) | <0.0001 |

Abbreviations: *ARDS, acute respiratory distress syndrome*;RR, respiratory rate; PEEP, positive end-expiratory pressure; $FiO_2$, inspired fraction of oxygen;$PaO_2$, partial pressure of oxygen; $PaO_2/FIO_2$, partial pressure of oxygen to fraction of inspired oxygen; $P_{plat}$, plateau pressure; VT, tidal volume; PIP, peak inspiratory pressure; DP, driving pressure;ECMO, extracorporeal membrane oxygenation; sd, standard deviation;IQR (interquartile range); ABG, Artery blood gas.

[a] *P* value represents comparisons between the survivors and non-survivors ARDS patients.

[b]*P* values for the comparison of continuous variables were calculated using the t-test.

driving pressure ≤14 $cmH_2O$ and a P/F ratio >150, the group with a driving pressure >14 $cmH_2O$ and a P/F ratio ≤150 exhibited a 2.497-fold higher risk of mortality (OR 2.497, 95% CI 1.201–5.191, P = 0.014) (Table 3). Patients with a driving pressure >14 $cmH_2O$ (Fig 2A, *p* = 0.025) and a P/F ratio ≤150 (Fig 2B, *p* = 0.002) had alower hospital survival rate. By combining these factors, patients with worse oxygenation and higher driving pressure had the lowest hospital survivalrate (Fig 2C, *p*<0.0001).

## Discussion

The present study was carried out in 6 ICUs of a referral medical center in central Taiwan. ARDS appears to represent an important health problem, which is both common and has a

**Table 3. Multivariate logistic regression analysis of factors associated with mortality in ARDS patients.**

| Variables | Univariable | | | Multivariable | | |
|---|---|---|---|---|---|---|
| | Odds ratio | 95%CI | P Value [a] | Odds ratio | 95%CI | P Value [b] |
| Prone | | | | | | |
| No | | 1[Reference] | | | | |
| Yes | 1.231 | (0.371–4.086) | 0.734 | | | |
| Aspiration of gastric contents | | | | | | |
| No | | 1[Reference] | | | 1[Reference] | |
| Yes | 2.058 | (0.821–5.161) | 0.124 | 1.996 | (0.870–4.578) | 0.103 |
| Charlson Comorbidity Index (CCI) | 1.110 | (1.008–1.222) | 0.034 | 1.072 | (0.985–1.166) | 0.106 |
| MAP, per 1 cmH$_2$O increment | 1.012 | (0.934–1.098) | 0.763 | | | |
| BMI, per 1 kg/m$^2$ increment | 0.969 | (0.912–1.029) | 0.303 | | | |
| APACHEII score | 1.091 | (1.041–1.145) | <0.0001 | 1.089 | (1.045–1.135) | <0.0001 |
| RR | 1.037 | (0.974–1.104) | 0.256 | | | |
| FiO$_2$ | 0.526 | (0.113–2.463) | 0.415 | | | |
| DP≤14 and P/F ratio>150 | | 1[Reference] | | | 1[Reference] | |
| DP≤14 and P/F ratio≤150 | 0.911 | (0.421–1.970) | 0.813 | 0.897 | (0.487–1.652) | 0.726 |
| DP>14 and P/F ratio>150 | 1.649 | (0.667–4.072) | 0.279 | 1.685 | (0.790–3.596) | 0.177 |
| DP>14 and P/F ratio≤150 | 2.995 | (1.175–7.635) | 0.022 | 2.497 | (1.201–5.191) | 0.014 |

Abbreviations: ARDS, acute respiratory distress syndrome; MAP, Mean Airway Pressure; BMI, Body Mass Index; APCHE II, Acute Physiology And Chronic Health Evaluation II; CCI, Charlson comorbidity index; CI., confidence interval;RR, respiratory rate;FiO$_2$, inspired fraction of oxygen;DP, Driving pressure; PaO$_2$/FiO$_2$, partial pressure of oxygen to fraction of inspired oxygen.

[a] Univariate analysis.

[b] All variables included in the multivariable analysis are reported in this Table.

high mortality, in critically ill patients receiving invasive mechanical ventilation. We found APACHE II scores and a combination of oxygenation and driving pressure were independent mortality risk factors. A combination of oxygenation status and respiratory mechanics gives better outcome prediction in ARDS patients. Adherenceto contemporary ventilation strategies and adjuncts in ARDS patients remains unsatisfactory. These findings also indicate the potential for improvement in the management of patients with ARDS.

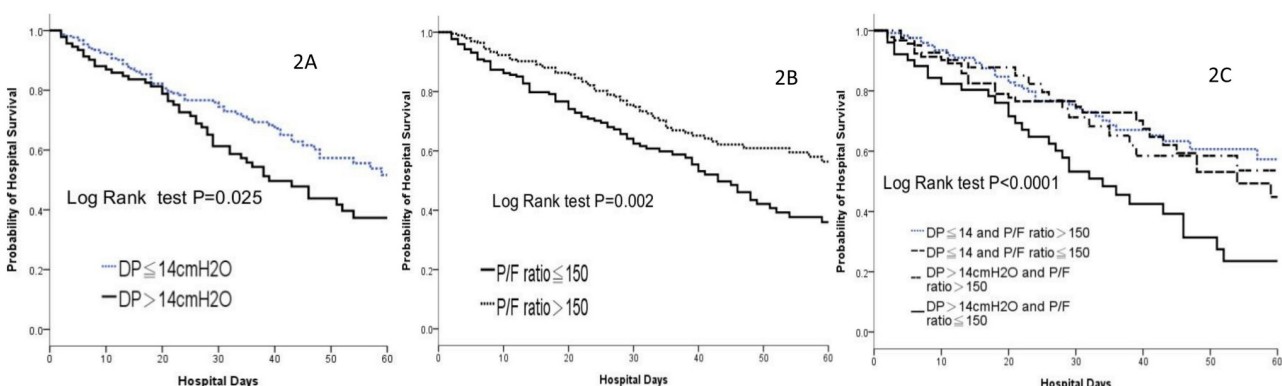

**Fig 2. Prediction of hospital survival when driving pressure and P/F ratio are combined.** (A) Probability of hospital survival by driving pressure. (B) Probability of hospital survival by P/Fratio. (C) Probability of hospital survival by combination of oxygen and driving pressure.DP, Driving pressure; P/F ratio, ratio of partial pressure of oxygen to fraction of inspired oxygen.

The burden of critical illness is increasing worldwide because of aging populations, natural disasters, conflicts, and higher-risk medical therapies [22]. The number of critically ill patients admitted to ICUs is increasing and most of these patients need mechanical ventilation [23]. In the LUNG SAFE study, approximately a quarter of patients were admitted to the ICU and who received mechanical ventilation developed ARDS [9], and there is a geographical variation in the prevalence of ARDS, which is probably due to the health care system, medical resource availability, and ethnics. Previous epidemiological studies of ARDS in Taiwan are limited due to the smallsize of case series [24] and the imprecise case definition used [25]. As there has been a substantial evolution in the health care system over the past two decades, there has been the development of a new ARDS definition [16], and epidemiological data on the prevalence, pattern of care, and outcome of ARDS patients in Taiwan are urgently needed. Our study demonstrates that the prevalenceof ARDS remains high, and that the mortality rate of these patients is doubled compared with patients without ARDS. This highlights the fact that critical care teams should pay more attention to the prevention and management of patients with, or at risk of ARDS.

In the real-life practice, recognition of ARDS is often delayed or missed, especially in those with mild disease. However, the mortality of ARDS patients remains high even in those with mild ARDS. In this cohort study of ARDS patients, the mortality rate of mild disease was 35.9%, which is similar to the LUNG SAFE study, but still much higher than those without ARDS. The mortality rate is even higher if ARDS worsens over the following days. Pham et al. emphasized the need to pay close attention to those with mild ARDS [26]. Failure to recognize ARDS in a timely fashion can lead to failure to implement strategies that improve survival of ARDS. Timely implementation of lung protection with low VT is fundamental to successful treatment in ARDS [27]. In severe influenza ARDS patients, first VT, shortly after intubation, is associated with increased mortality [28]. In this study, although ARDS was recognized in a timely manner at the ICU admission, the outcomes of these patients were not satisfactory. This can be attributed to adherence to lung protection and the application of adjunct therapies. Another important issue for ARDS in real-life practice is that it is often under-recognized. The LUNG SAFE study also indicated that a lower ratio of healthcare professionals to ICU patients was associated with reduced recognition of ARDS. A global epidemiological investigation into the burden of critical illness also suggested that ICU organization has an important effect on the risk of death [29]. Taken together, our results also highlight the importance of timely identification and proper management of ARDS, both of which may be improved by investing resources to improve the ICU staffing and resources.

As hypoxemia is the major clinical presentation of ARDS, the severity of ARDS is traditionally classified by the P/F ratio [30]. As variations in PEEP and $FiO_2$ levels can impact the P/F ratio, it is now considered that PEEP or continuous positive airway pressure $\geq 5$ cmH$_2$O is required to define and classify ARDS severity. The difference in mortality between mild and severe ARDS was around 10% in the LUNG SAFE and our study. Although our study did not reach statistical significance, the mortality rates were similar to those of the LUNG SAFE study (Mild: 35.9; Severe: 46.5; difference: 10.6%).Respiratory mechanics, such as $P_{plat}$ [31] and driving pressure [19], are good predictors of outcomes in ARDS patients [32]. In addition, improvement of respiratory mechanics, such as an increase in dynamic driving pressure [33] or a decrease in $PaCO_2$ [34], are associated with improved survival in prone positioning of severe ARDS patients. The respiratory response to prone positioning was more relevant when $PaCO_2$ rather than the P/F ratio was used [35]. In this study, by combining respiratory mechanics and oxygenation status, we found that patients with a driving pressure >14 cmH$_2$O and a P/F $\leq 150$ have the worst outcomes. ARDS is also an inflammatory disease, both locally and systemically, and hyper inflammation is associated with clinical outcomes. We therefore

suggest the need to develop a new classification model which combines oxygenation, respiratory mechanics and inflammation parameters and which may better predict outcomes for ARDS patients.

A major strength of this study was that the data were collected retrospectively from a quality improvement program for a year. In contrast to previous studies which were conducted over 4 weeks (the LUNG SAFE) to 2 months [36], this epidemiological study avoided any seasonal variation in ARDS prevalence. This study also had several limitations. First, this was a single center study and is inherently limited for external validation. As the Taiwan National Health Insurance is a single payer system, the hospitals share high homogeneity in their practice patterns and ICU organizations. As this was an observational study without intervention, the results of this study are representative of real-life information about the diagnosis and management of ARDS in the ICUs. Second, we only included patients with invasive mechanical ventilation within 24 hours of the ICU admission. Patients with non-invasive ventilation were not included. These patients either had mild ARDS with a good prognosis or were those designated as 'do not intubate' due to their advanced age or end-stage disease. We also did not include ARDS which developed after the first day of the ICU mission because most ARDS cases wereearly onset in the ICU admission. Thirdly, there is a lack of information regarding pharmacological treatments for ARDS. Currently, there is no established medication for ARDS that can reduce mortality rates in either the short or long term. Although several clinical trials have been conducted to explore various pharmacological treatments with the aim of improving clinical outcomes for ARDS, none have demonstrated effectiveness. In our study, we primarily focused on the epidemiological aspects of ARDS and the impact of risk factors on mortality outcomes. Therefore, we did not delve deeply into the exploration of pharmacotherapy. However, all these limitations should be taken into account when interpreting the information collected.

## Conclusions

ARDS is common among critically ill patients who are admitted to the ICU and who use invasive mechanical ventilation. Hospital mortality remains high and could potentially be improved by adherence to currently available evidence. Combining oxygenation status and respiratory mechanics may better predict ARDS patients at risk of mortality.

## Supporting information

**S1 Table. Characteristics of ARDS patients categorized by severity.**
(DOCX)

**S2 Table. Mechanical ventilation and arterial blood gas on admission by ARDS severity.**
(DOCX)

**S3 Table. Characteristics of ARDS patients between DP≦14 and DP>14.**
(DOCX)

**S4 Table. Mechanical ventilation and adjunctive therapy between DP≦14 andDP>14.**
(DOCX)

**S1 Dataset.**
(XLSX)

## Acknowledgments

We thank the Biostatistics Group, Department of Medical Research, TaichungVeterans General Hospital for their assistance in statistical consultation.

## Author Contributions

**Conceptualization:** Yu-Yi Yu, Chieh-Laing Wu, Ming-Cheng Chan.

**Data curation:** Yu-Yi Yu, Wei-Fan Ou, Jia-Jun Wu, Chieh-Laing Wu.

**Formal analysis:** Yu-Yi Yu, Wei-Fan Ou, Jia-Jun Wu.

**Investigation:** Yu-Yi Yu, Wei-Fan Ou, Jia-Jun Wu, Ming-Cheng Chan.

**Methodology:** Yu-Yi Yu, Wei-Fan Ou, Jia-Jun Wu, Ming-Cheng Chan.

**Project administration:** Yu-Yi Yu, Ming-Cheng Chan.

**Resources:** Yu-Yi Yu.

**Software:** Yu-Yi Yu.

**Supervision:** Chieh-Laing Wu, Ming-Cheng Chan.

**Validation:** Han-Shui Hsu, Kuang-Yao Yang, Ming-Cheng Chan.

**Writing – original draft:** Yu-Yi Yu, Ming-Cheng Chan.

**Writing – review & editing:** Han-Shui Hsu, Kuang-Yao Yang, Ming-Cheng Chan.

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
