## [Decision Letter · Decision Letter 0]

17 Jul 2023

PONE-D-23-11371A combination of oxygenation and driving pressure can provide valuable information in predicting the risk of mortality in ARDS patients.PLOS ONE

Dear Dr. Chan,

Thank you for submitting your manuscript to PLOS ONE. After careful consideration, we feel that it has merit but does not fully meet PLOS ONE’s publication criteria as it currently stands. Therefore, we invite you to submit a revised version of the manuscript that addresses the points raised during the review process.

 Please submit your revised manuscript by Aug 31 2023 11:59PM. If you will need more time than this to complete your revisions, please reply to this message or contact the journal office at plosone@plos.org. Please include the following items when submitting your revised manuscript:A rebuttal letter that responds to each point raised by the academic editor and reviewer(s). You should upload this letter as a separate file labeled 'Response to Reviewers'.A marked-up copy of your manuscript that highlights changes made to the original version. You should upload this as a separate file labeled 'Revised Manuscript with Track Changes'.An unmarked version of your revised paper without tracked changes. You should upload this as a separate file labeled 'Manuscript'.

We look forward to receiving your revised manuscript.

Kind regards,

Academic Editor

PLOS ONE

Journal Requirements:

"This study was supported by Taichung Veterans General Hospital (TCVGH-1104101C). The funders had no role in study design, data collection and analysis, decision to publish, or preparation of the manuscript."

We note that one or more of the authors is affiliated with the funding organization, indicating the funder may have had some role in the design, data collection, analysis or preparation of your manuscript for publication; in other words, the funder played an indirect role through the participation of the co-authors. If the funding organization did not play a role in the study design, data collection and analysis, decision to publish, or preparation of the manuscript and only provided financial support in the form of authors' salaries and/or research materials, please do the following:

(1) Review your statements relating to the author contributions, and ensure you have specifically and accurately indicated the role(s) that these authors had in your study. These amendments should be made in the online form.

(2) Confirm in your cover letter that you agree with the following statement, and we will change the online submission form on your behalf: 

**Additional Editor Comments:**

Please revise.

Reviewers' comments:

Reviewer's Responses to Questions

**Comments to the Author**

1. Is the manuscript technically sound, and do the data support the conclusions?

Reviewer #1: No

Reviewer #2: Partly

Reviewer #3: Yes

2. Has the statistical analysis been performed appropriately and rigorously? 

Reviewer #1: No

Reviewer #2: No

Reviewer #3: Yes

3. Have the authors made all data underlying the findings in their manuscript fully available?

Reviewer #1: Yes

Reviewer #2: Yes

Reviewer #3: Yes

4. Is the manuscript presented in an intelligible fashion and written in standard English?

Reviewer #1: Yes

Reviewer #2: Yes

Reviewer #3: Yes

5. Review Comments to the Author

Reviewer #1: A combination of oxygenation and driving pressure can provide valuable information in predicting the risk of mortality in ARDS patients.

PONE-D-23-11371

This is a secondary analysis of prospectively collected data of a clinical audit, aiming to describe a cohort of ARDS patients. Obviously, so much effort and time was invested in this study that addresses an important topic in critical care. I would like to thank the authors for their submission, however, I have several comments and concerns:

1- The most important concern is that in several sections of the article you described oxygenation (P/F ratio) as an independent risk factor of mortality, and this was what you depended on to combine driving pressure and oxygenation in the Kaplan Meier curves. However, your results do not support that claim, P/F ratio was not significantly different between survivors and deceased (table 2) and it was not included in the logistic regression model. So, how did you reach the conclusion that it is an independent risk factor of mortality?

2- Abstract: you wrote (The objective of this study was to investigate the incidence and outcomes of ARDS): Actually what was described (370 ARDS / 1778 ventilated patients) is prevalence rather than incidence, because the patients already had ARDS, incidence is when patients newly develop the condition that they didn’t have before.

3- The total number of screened patients is 1788 in the abstract, but 1778 in the results. Please verify.

4- Abstract: For driving pressure, If OR is 1.10, then 1 cmH2O increase is associated with 10% (not 11%) increase in odds of death (not in mortality rate).

5- Introduction: The first 7 lines need references.

6- Method: Please define early in the method at least 1 primary objective and a few secondary ones.

7- Why is the age of adulthood considered as above 20 years? This is uncommon, is this the routine in your institute?

8- I have several comments about the statistical plan:

a- Your analysis includes multiple testing and extensive analysis. So, please indicate whether or not correction for multiple testing was performed. If not, then you should indicate that the results should be interpreted with caution or that they are not conclusive.

b- One way ANOVA compares means of continuous variables among 3 or more groups, so, how did you compare categorical variables between the 3 groups of ARDS severity? I assume Pearson’s Chi Square, but it should be mentioned.

c- You wrote (Kaplan-Meier analysis was performed to test the mortality correlation of driving pressure). Kaplan Meier curves and Log Rank tests are not used to test correlation, they compare survival rates.

d- The logistic regression model needs clarification: How were the included variables chosen? How did you build the multi-variable logistic regression model? Particularly since there were other variables that were significantly different between survivors and deceased (FiO2 - RR - PIP - Pplat - Compliance) that you left out? Usually, we either perform uni-variable logistic regression and then include any variable with p value < 0.1 or 0.15 in the multi-variable model. Alternatively, we may perform logistic regression with backward or forward elimination to retain variables in the model with p value < a predefined cutoff value (usually 0.1). I strongly advise you to consult a biostatistician.

9- Results: Frequently you use the description (paralleled to severity), while almost always numerically correct, sometimes this parallel change is not statistically significant (such as for driving pressure). Please differentiate when there is statistical significance and when there is not.

10- SOFA scores did not parallel severity, it was lower for severe ARDS on days 3 and 7.

11- Results: You wrote (There was no significant difference in age, gender, BMI, comorbidities, admission source, type of ICU, etiology,…..). Actually, type of ICU was significantly different.

12- Results: You wrote: (As oxygenation (P/F ratio) and respiratory mechanics (driving pressure) were both associated with mortality in ARDS). I commented on this before. There is no evidence that P/F ratio is associated with mortality in your results.

13- Discussion: In general, it needs to be shortened, and more focused. Do not astray from the main topic to involve issues such as DNR.

14- The results of the Kaplan Meier curves can’t be interpreted as predictors, accordingly, I don’t agree with your conclusion that combining driving pressure and oxygenation predicts outcome. This could have been true if that combined variable was incorporated in a predictive model (such as logistic regression).

15- LUNG SAFE is reference number 21 not 4.

16- There are other limitations, for example, the lack of information about pharmacological treatment, non-robust logistic regression model (unless you improve it).

17- For tables 1 and 2: Please indicate in the footnotes when t-test was used and when Mann Whitney was used.

18- Finally, please proofread the manuscript for minor errors of grammar and spelling.

Reviewer #2: This looks a good article to see the prognostic factors in ARDS. Driving pressure has been studied extensively in recent years. I have a few suggestions

1. Abstract- Objective and title do not match. Diagnosis of 1788 patients in ICUs is not clear but explained in results.

2. Is it prospective or retrospective study?

3. About 85% have record of driving pressure but in table all 370 patient have been analyzed

A few suggestions have been added in the manuscript.

I hope you will consider.

Reviewer #3: dear authors

thank you for this valuable work, please take the following comments into your consideration.

1-in the introduction section: omit the word " non-pulmonatry " and rewrite the sentence (sepsis ,trauma ,....)

2-add references for the first and second paragraph of the introduction

3-it is written in the abstract section date was collected prospectively, while in the methods section it is written retrospectively? please explain

4-the same in the discussion section: you wrote the data was collected prospectively, please explain this

5-some results were repeated in the discussion section, this should be avoided.

6. PLOS authors have the option to publish the peer review history of their article (what does this mean?). If published, this will include your full peer review and any attached files.

Reviewer #1: No

Reviewer #2: No

Reviewer #3: No

---

## [Author Response · Author response to Decision Letter 0]

30 Aug 2023

We would like to thank the reviewers for their extensive assessment of our manuscript, and for important and helpful comments and suggestions. We have taken all the remarks into account, in a manner that is described in detail below together with our answers to certain comments. We have responded to all the reviewer’s comments in a point-by-point manner and have revised the manuscript accordingly. The revised portions are indicated by red font. We think that, following the reviewers’ suggestions, our manuscript has gained in clarity and hope that the changes made will be considered satisfactory.

Responses to Reviewer #1

Comment 1-1:

The most important concern is that in several sections of the article you described oxygenation (P/F ratio) as an independent risk factor of mortality, and this was what you depended on to combine driving pressure and oxygenation in the Kaplan Meier curves. However, your results do not support that claim, P/F ratio was not significantly different between survivors and deceased (table 2) and it was not included in the logistic regression model. So, how did you reach the conclusion that it is an independent risk factor of mortality?

Response:

We appreciate the reviewer's comments regarding this important issue and the need for further clarification. The primary clinical manifestation of Acute Respiratory Distress Syndrome (ARDS) is hypoxemia. Traditionally, the severity of ARDS has been categorized using the PaO2/FiO2 ratio, and it`s related to mortality. The Lung Safe study reported a difference in mortality rates of around 10% between mild and severe patients of ARDS. In our study, we observed a similar difference in ARDS mortality rates (Mild 35.9%; Severe 46.5%, difference 10.6%). Due to the relatively small sample size in our study, this difference did not reach statistical significance. In order to better identify the risk factors associated with mortality in ARDS patients, we conducted an analysis by combining the P/F ratio with the driving pressure. 

 We have revised the statement in the discussion section, and the narrative is as follows: The difference in mortality between mild and severe ARDS was around 10% in the LUNG SAFE and our study. Although our study did not reach statistical significance, the mortality rates were similar to those of the LUNG SAFE study (Mild: 35.9%; Severe: 46.5%; difference: 10.6%) (Page 20-21, line 331-334). 

Comment 1-2:

Abstract: you wrote (The objective of this study was to investigate the incidence and outcomes of ARDS): Actually what was described (370 ARDS / 1778 ventilated patients) is prevalence rather than incidence, because the patients already had ARDS, incidence is when patients newly develop the condition that they didn’t have before.

Response:

We appreciate the reviewer's attention to detail .In response to the reviewer's viewpoint, we have revised the statements in the manuscript to accurately reflect the prevalence of ARDS in our study population. We have provided a statement to clarify this issue.

 The revised statement is as follows:

1、 A total of 1,778 subjects were screened in 6 adult ICUs and 370 patients fulfilled the criteria of ARDS in the first 24 hours of the ICU admission. Among these patients, the prevalence of ARDS was 20.8% and the overall hospital mortality rate was 42.2%. (page 5, lines78-81)

2、 In Taiwan, the epidemiological data, patterns of care and outcomes of patients with ARDS are not clear. Given that ARDS has an important impact on public health burden, we conducted a retrospective observational study to investigate its prevalence and factors associated with the outcomes of critically ill patients with ARDS. (page 8, lines114-118)

3、 In the LUNG SAFE study, approximately a quarter of patients were admitted to the ICU and who received mechanical ventilation developed ARDS [9], and there is a geographical variation in the prevalence of ARDS, which is probably due to the health care system, medical resource availability, and ethnics. (page18,line 291-295)

4、 As there has been a substantial evolution in the health care system over the past two decades, there has been the development of a new ARDS definition [16], and epidemiological data on the prevalence, pattern of care, and outcome of ARDS patients in Taiwan are urgently needed. Our study demonstrates that the prevalence of ARDS remains high, and that the mortality rate of these patients is doubled compared with patients without ARDS. (Page 19, lines297-302).

5、 In contrast to previous studies which were conducted over 4 weeks (the LUNG SAFE) to 2 months [36], this epidemiological study avoided any seasonal variation in ARDS prevalence. (Page 21, lines348-350)

 Comment 1-3:

The total number of screened patients is 1788 in the abstract, but 1778 in the results. Please verify.

Response:

We appreciate the reviewer for bringing this to our attention. Upon careful reevaluation, the correct total number of screened patients is 1,778, as reported in the abstract section. The number provided in the abstract was a typographical error, and we have corrected it accordingly (page 5, lines 78). 

 Comment 1-4:

Abstract: For driving pressure, If OR is 1.10, then 1 cmH2O increase is associated with 10% (not 11%) increase in odds of death (not in mortality rate).

Response:

We thank the reviewer for highlighting this significant issue. We have revised the driving pressure narrative to state that an increase of 1cmH2O increase is associated with a 10% increase in odds of death (page 5, lines 82-84).

Comment 1-5:

Introduction: The first 7 lines need references.

Response:

We thank the reviewer for the valuable reminder. We have added the new references for the first 7 lines of the introduction. Specifically we have cited the new reference [1, 2] to support our statements (page 7 , lines 94-96).Additionally we have cited the new reference[2-4] to support our statements (page 7, lines 96-99), and the new reference[1, 5]to support our statements (page 7, lines 99-100). The first seven lines of the revised introduction are provided below:

Acute respiratory distress syndrome (ARDS) is a life-threatening condition which may result from a variety of pulmonary (e.g., pneumonia and aspiration) and extra-pulmonary causes (e.g., non-pulmonary sepsis, trauma and pancreatitis)[1, 2]. ARDS is characterized by acute, diffuse alveolar inflammation and flooding, leading to increased capillary permeability, which results in lung tissue edema and loss of aeration[2-4]. Clinically, patients with ARDS often present with hypoxemia, pulmonary congestion, and decreased respiratory compliance[1, 5]

Comment 1-6:

Method: Please define early in the method at least 1 primary objective and a few secondary ones.

Response:

We appreciate the reviewer's valuable feedback, which has enabled us to enhance the clarity and relevance of our research objectives.

In response to the reviewer's suggestion, we have included the primary objective and a secondary objective in the method section of our revised manuscript (page 9, lines 142-145).

The revised statement is as follows: 

Types of Outcome Measures

Our primary outcome was to investigate factors that impact ARDS mortality. 

The secondary outcome was to understand the real-life practice pattern in management of ARDS in Taiwan.

Comment 1-7:

Why is the age of adulthood considered as above 20 years? This is uncommon, is this the routine in your institute?

Response:

We thank the reviewer for allowing us to clarify this important issue. In Taiwan, the legal age of adulthood was 20 years old. However, on January 1st, 2023, the Civil Law was amended to redefine the legal age of adulthood as 18 years old. The enrollment period for our study was from 2018 to 2019. To adhere to the local law, we enrolled patients as 20 years old and above.

Comment 1-8:

I have several comments about the statistical plan:

a- Your analysis includes multiple testing and extensive analysis. So, please indicate whether or not correction for multiple testing was performed. If not, then you should indicate that the results should be interpreted with caution or that they are not conclusive.

Response:

We appreciate the reviewer for highlighting this significant concern. Our predictive model specifically focuses on mortality as the primary outcome, with the main predictive factor being the driving pressure. There were no considerations raised regarding the issue of multiple testing.

b- One way ANOVA compares means of continuous variables among 3 or more groups, so, how did you compare categorical variables between the 3 groups of ARDS severity? I assume Pearson’s Chi Square, but it should be mentioned.

Response:

We thank for the reviewer for reminding us this important issue. We utilized Pearson's Chi Square test to compare categorical variables among the three groups of ARDS severity and addressed it in the manuscript's Statistical analysis section (page 11, lines 176-178). The statements read as: One-way analysis of variance was used to compare variables among the three ARDS severity groups and the Pearson's Chi Square test to compare categorical variables among the three groups of ARDS severity.

c- You wrote (Kaplan-Meier analysis was performed to test the mortality correlation of driving pressure). Kaplan Meier curves and Log Rank tests are not used to test correlation, they compare survival rates.

Response:

We appreciate the reviewer for giving us the opportunity to provide further clarification on this issue. We completely agree with the reviewer's viewpoint that Kaplan Meier curves and Log Rank tests are used to compare survival rates. We have addressed this in the manuscript's Statistical analysis section (page 11, lines 179-180). The revised statements are as follows: Kaplan-Meier analysis was performed to test the mortality of driving pressure above or below 14 cmH2O in different ARDS severities.

d- The logistic regression model needs clarification: How were the included variables chosen? How did you build the multi-variable logistic regression model? Particularly since there were other variables that were significantly different between survivors and deceased (FiO2 - RR - PIP - Pplat - Compliance) that you left out? Usually, we either perform uni-variable logistic regression and then include any variable with p value < 0.1 or 0.15 in the multi-variable model. Alternatively, we may perform logistic regression with backward or forward elimination to retain variables in the model with p value < a predefined cutoff value (usually 0.1). I strongly advise you to consult a biostatistician.

Response:

 We thank the reviewer for raising these important points in the statistical methods. To address this concern, we have consulted with a biostatistician to reevaluate the variable selection process. We carefully assessed the clinical relevance of each variable and their significance in predicting the outcome.

In our study, the selection of variables for inclusion in the logistic regression model was based on a combination of clinical relevance and statistical significance. In response to the reviewer's suggestion, any variable with a p-value ≤ 0.1 in the univariable analysis was selected into multivariable analysis. We have included the FiO2 and RR variables in the univariable logistic regression analysis (as shown in Table 3 and S5 Table). 

The reason for not including the PIP, Pplat, and Compliance variables was due to their high correlation and collinearity with the driving pressure.

Comment 1-9:

Results: Frequently you use the description (paralleled to severity), while almost always numerically correct, sometimes this parallel change is not statistically significant (such as for driving pressure). Please differentiate when there is statistical significance and when there is not.

Response:

 We appreciate the reviewer's attention to detail and the opportunity they have given us to address the narrative. In our manuscript, we state that the measured pressures, including PIP, Pplat and driving pressure changed in parallel with ARDS severity. Although there was no statistically significant difference in the driving pressure, as shown in S2 table.

We have rewritten the statement as follows: Volume-targeted ventilation (82.4%) was preferred in this cohort study of patients. Regarding the ventilator settings, FiO2, PEEP and respiratory rate increased with ARDS severity, whereas VT decreased. The measured pressures, including PIP and Pplat changed in parallel with ARDS severity (page 13 and 14, lines 216-219).

Comment 1-10:

SOFA scores did not parallel severity, it was lower for severe ARDS on days 3 and 7.

Response:

 We appreciate the reviewer's attention to detail and giving us the opportunity to provide more explanation regarding this issue. Due to some patient deaths that impacted the data collection of SOFA scores, we have decided to exclude the scores for day 3 and day 7 in order to avoid confusion (as shown in S2 table).

 Comment 1-11:

 Results: You wrote (There was no significant difference in age, gender, BMI, comorbidities, admission source, type of ICU, etiology,…..). Actually, type of ICU was significantly different.

Response:

We appreciate the reviewer's attention to detail and have thoroughly rechecked the data in Table 1, S1Table and S3 table. After carefully reviewing the data, we can confirm that there was no significant difference the type of ICUs between in table 1 and S1 table, but there was a significant difference in S3 table. We have addressed the statement in the manuscript's Characteristics of ARDS patients according to driving pressure section (page 15, lines 239-242). These statements read as: There were no significant difference in age, gender, BMI, co-morbidities, admission source, etiology, length of ICU stay, length of hospital stay or ventilator days between groups of driving pressure (DP) ≤14cmH2O and DP >14 cmH2O.

 Comment 1-12:

Results: You wrote: (As oxygenation (P/F ratio) and respiratory mechanics (driving pressure) were both associated with mortality in ARDS). I commented on this before. There is no evidence that P/F ratio is associated with mortality in your results.

Response:

 We appreciate the reviewer for giving us the opportunity to provide more explanation regarding this issue. As hypoxemia is the major clinical presentation of ARDS, the severity of ARDS is traditionally classified by the P/F ratio. The Lung Safe study highlighted a mortality rate discrepancy of approximately 10% between mild and severe ARDS. In our study, we observed a similar variation in ARDS mortality rates (Mild 35.9%; Severe 46.5%, difference 10.6%). Due to the relatively small sample size in our study, this difference did not reach statistical significance. In order to better identify the risk factors associated with mortality in ARDS patients, we conducted an analysis by combining the P/F ratio with the driving pressure.

 We have revised the statement in the discussion section, and the narrative is as follows: The difference in mortality between mild and severe ARDS was around 10% in the LUNG SAFE and our study. Although our study did not reach statistical significance, the mortality rates were similar to those of the LUNG SAFE study (Mild: 35.9%; Severe: 46.5%; difference: 10.6%) (page20-21, lines 331-334).

 Comment 1-13:

Discussion: In general, it needs to be shortened, and more focused. Do not astray from the main topic to involve issues such as DNR.

Response:

 We would like to thank the reviewers for their feedback of our manuscript. In response to the reviewer’s suggestion, we have revised the statement for the discussion section. We have removed the original narrative from our manuscript “The concept of limiting life-sustaining therapies or measures in critically ill patients varies among Asian countries and regions and Co-morbidities are not modifiable and can also serve as a key factor in decisions to forgo life-sustaining treatment.”

 Due to the revised logistic regression model, the CCI was not found to be significant in the multivariate logistic regression analysis. As a result, we have decided to exclude the co-morbidity statement from the discussion section. 

Comment 1-14:

The results of the Kaplan Meier curves can’t be interpreted as predictors, accordingly, I don’t agree with your conclusion that combining driving pressure and oxygenation predicts outcome. This could have been true if that combined variable was incorporated in a predictive model (such as logistic regression).

Response:

 We fully agree with the reviewer's suggestion and have revised this issue accordingly. The Kaplan-Meier curves and Log-Rank tests are used to compare survival rates. In response to the reviewer’s suggestion, we have conducted logistic regression to explore the relationship between the P/F ratio and driving pressure. Please refer to the S5 table for the details.

Therefore, combining these factors allows us to better assess ARDS patients' outcomes, and these results are also compatible with the clinical presentation.

Comment 1-15:

LUNG SAFE is reference number 21 not 4.

Response:

We thank the reviewer for the clarification. In response to the reviewer’s suggestion, we have provided further explanations for these issues in the reference section. Due to the revision of reference citations in the introduction, there have been changes to the reference numbers. The original reference number 4 has been updated to 9, and the original reference number 21 has been updated to 32.

Epidemiology, Patterns of Care, and Mortality for Patients with Acute Respiratory Distress Syndrome in Intensive Care Units in 50 Countries. JAMA, 2016; 315(8): 788-800. (page26, line442-446)

Potentially modifiable factors contributing to outcome from acute respiratory distress syndrome: the LUNG SAFE study. Intensive Care Med, 2016; 42(12): 1865-1876. (page29, line540-543)

 Together, these two articles provide valuable information on both the epidemiological aspects and potentially modifiable factors affecting the outcome of ARDS in ICUs. The LUNG SAFE study as a whole has significantly contributed to the understanding and management of ARDS, ultimately benefiting patients in critical care settings worldwide.

 Comment 1-16:

There are other limitations, for example, the lack of information about pharmacological treatment, non-robust logistic regression model (unless you improve it).

Response:

We thank the reviewer for bringing these limitations to our attention. In response to the reviewer's suggestion, we have added the narrative to the limitation section. The statement is as follows: Thirdly, there is a lack of information regarding pharmacological treatments for ARDS. Currently, there is no established medication for ARDS that can reduce mortality rates in either the short or long term. Although several clinical trials have been conducted to explore various pharmacological treatments with the aim of improving clinical outcomes for ARDS, none have demonstrated effectiveness. In our study, we primarily focused on the epidemiological aspects of ARDS and the impact of risk factors on mortality outcomes. Therefore, we did not delve deeply into the exploration of pharmacotherapy (page 22, lines 362-370).

In order to address this concern and improve the statistical analysis, we have consulted with biostatisticians. Base on their suggestion, we have revised our logistic regression model to provide more reliable insights into the factors impacting ARDS mortality outcomes.

Comment 1-17:

For tables 1 and 2: Please indicate in the footnotes when t-test was used and when Mann Whitney was used.

Response:

We thank the reviewer for bringing this detail to our attention. In tables 1 and 2, we have included footnotes indicating the statistical methods used for analysis, which include the t-test and Mann-Whitney U test (page 33-35).

Comment 1-18:

Finally, please proofread the manuscript for minor errors of grammar and spelling.

Response:

 We appreciate the reviewer for reminding us of this important issue. After carefully reviewing the entire manuscript attentively, we have meticulously checked every word in the manuscript for correct spelling.

Responses to Reviewer #2

Comment 2-1:

Abstract- Objective and title do not match. Diagnosis of 1788 patients in ICUs is not clear but explained in results.

Response:

In response to the reviewer’s suggestion, we have revised our objective to align with the title. The diagnosis of 1,778 subjects was conducted in 6 adult ICUs, and this clarification has been added to the abstract.

We have revised the statements in the abstract section, including the background and results. The narrative is as follows:

(Background) The objective of this study was to investigate the risk factors that impact ARDS mortality in a medical center in Taiwan (page5, lines 71-72).

 (Results) A total of 1,778 subjects were screened in 6 adult ICUs and 370 patients fulfilled the criteria of ARDS in the first 24 hours of the ICU admission (page 5, lines 78-79).

Comment 2-2:

Is it prospective or retrospective study?

Response:

We thank the reviewer for allowing us to further explain this issue. This is retrospective observational study.

Comment 2-3:

About 85% have record of driving pressure but in table all 370 patient have been analyzed

Response:

 We thank the reviewer for allowing us to further explain this issue. The total number of subjects was 370; driving pressure measurements were taken only for 318 of them. The statement in result section is as follows: We further analyzed the characteristics of patients by driving pressure. Driving pressure at admission was available in 318 (85.9%) of the patients (page15, lines 236-237).

Responses to Reviewer #3

Comment 3-1:

in the introduction section: omit the word " non-pulmonatry " and rewrite the sentence (sepsis ,trauma ,....)

Response:

 We thank the reviewer for the suggestion. We have omitted the word" non-pulmonary” and rewritten the sentence (page 7, lines 94-96). The revised statement is as follows: Acute respiratory distress syndrome (ARDS) is a life-threatening condition which may result from a variety of pulmonary (e.g., pneumonia and aspiration) and extra-pulmonary causes (e.g., sepsis, trauma and pancreatitis)[1, 2].

Comment 3-2:

add references for the first and second paragraph of the introduction

Response:

 We thank the reviewer for reminding us of this important issue. We have been added the references in the first and second paragraph of the introduction.

The revised introduction are provided below:

Acute respiratory distress syndrome (ARDS) is a life-threatening condition which may result from a variety of pulmonary (e.g., pneumonia and aspiration) and extra-pulmonary causes (e.g., non-pulmonary sepsis, trauma and pancreatitis)[1, 2]. ARDS is characterized by acute, diffuse alveolar inflammation and flooding, leading to increased capillary permeability, which results in lung tissue edema and loss of aeration[2-4]. Clinically, patients with ARDS often present with hypoxemia, pulmonary congestion, and decreased respiratory compliance[1, 5] (page 7, line 94-100)

Comment 3-3:

it is written in the abstract section date was collected prospectively, while in the methods section it is written retrospectively? please explain

Response:

We appreciate the reviewer for allowing us to clarify this important issue. This is a retrospective study, some of the data obtained through a prospective ICU quality improvement program. In order not to confuse in the readers, we clarify that is a retrospective study. We have amended this important issue in abstract, introduction, method and discussion section .The statement is as follows:

1. This was a single center, observational study that retrospectively analyzed data from adults in 6 intensive care units (ICUs) at Taichung Veterans General Hospital in Taiwan from 1st October, 2018 to 30th September, 2019 (abstract section, page5, lines 73-75)

2. In Taiwan, the epidemiological data, patterns of care and outcomes of patients with ARDS are not clear. Given that ARDS has an important impact on public health burden, we conducted a retrospective observational study to investigate its prevalence and factors associated with the outcomes of critically ill patients with ARDS (introduction section, page8, lines 114-118). 

3. During the study period, we retrospectively collected data from a quality improvement program via a clinical audit of daily practice in patients with acute respiratory failure who needed mechanical ventilation in the ICUs. (method section, page9, lines 130-133)

4. A major strength of this study was that the data were collected retrospectively from a quality improvement program for a year. (discussion section, page 21, lines 347-348)

Comment 3-4:

the same in the discussion section: you wrote the data was collected prospectively, please explain this

Response:

We appreciate the reviewer for allowing us to clarify this important issue. This is a retrospective study, some of the data obtained through a prospective ICU quality improvement program. In order not to confuse in the readers, we clarify that is a retrospective study. We have amended this important issue in abstract, introduction, method and discussion section .The statement is as follows:

1. This was a single center, observational study that retrospectively analyzed data from adults in 6 intensive care units (ICUs) at Taichung Veterans General Hospital in Taiwan from 1st October, 2018 to 30th September, 2019 (abstract section, page5, lines 73-75). 

2. In Taiwan, the epidemiological data, patterns of care and outcomes of patients with ARDS are not clear. Given that ARDS has an important impact on public health burden, we conducted a retrospective observational study to investigate its prevalence and factors associated with the outcomes of critically ill patients with ARDS (introduction section, page 8, lines 114-118). 

3. During the study period, we retrospectively collected data from a quality improvement program via a clinical audit of daily practice in patients with acute respiratory failure who needed mechanical ventilation in the ICUs (method section, page 9, lines 130-133). 

4. A major strength of this study was that the data were collected retrospectively from a quality improvement program for a year (discussion section, page21, lines347-348). 

Comment 3-5:

some results were repeated in the discussion section, this should be avoided

Response:

 We thank the reviewer for reminding us of this important issue. In response to the reviewer’s suggestion, we have revised the discussion section by omitting the co-morbidity discussion.

---

## [Decision Letter · Decision Letter 1]

2 Nov 2023

PONE-D-23-11371R1A combination of oxygenation and driving pressure can provide valuable information in predicting the risk of mortality in ARDS patients.PLOS ONE

Dear Dr. Chan,

Thank you for submitting your manuscript to PLOS ONE. After careful consideration, we feel that it has merit but does not fully meet PLOS ONE’s publication criteria as it currently stands. Therefore, we invite you to submit a revised version of the manuscript that addresses the points raised during the review process. Please revise. Please submit your revised manuscript by Dec 17 2023 11:59PM. If you will need more time than this to complete your revisions, please reply to this message or contact the journal office at plosone@plos.org. Please include the following items when submitting your revised manuscript:A rebuttal letter that responds to each point raised by the academic editor and reviewer(s). You should upload this letter as a separate file labeled 'Response to Reviewers'.A marked-up copy of your manuscript that highlights changes made to the original version. You should upload this as a separate file labeled 'Revised Manuscript with Track Changes'.An unmarked version of your revised paper without tracked changes. You should upload this as a separate file labeled 'Manuscript'.If applicable, we recommend that you deposit your laboratory protocols in protocols.io to enhance the reproducibility of your results. Protocols.io assigns your protocol its own identifier (DOI) so that it can be cited independently in the future. For instructions see: https://journals.plos.org/plosone/s/submission-guidelines#loc-laboratory-protocols. Additionally, PLOS ONE offers an option for publishing peer-reviewed Lab Protocol articles, which describe protocols hosted on protocols.io. Read more information on sharing protocols at https://plos.org/protocols?utm_medium=editorial-email&utm_source=authorletters&utm_campaign=protocols.

We look forward to receiving your revised manuscript.

Kind regards,

Academic Editor

PLOS ONE

Journal Requirements:

Reviewers' comments:

Reviewer's Responses to Questions

**Comments to the Author**

1. If the authors have adequately addressed your comments raised in a previous round of review and you feel that this manuscript is now acceptable for publication, you may indicate that here to bypass the “Comments to the Author” section, enter your conflict of interest statement in the “Confidential to Editor” section, and submit your "Accept" recommendation.

Reviewer #1: (No Response)

Reviewer #3: All comments have been addressed

2. Is the manuscript technically sound, and do the data support the conclusions?

Reviewer #1: Partly

Reviewer #3: Yes

3. Has the statistical analysis been performed appropriately and rigorously? 

Reviewer #1: Yes

Reviewer #3: Yes

4. Have the authors made all data underlying the findings in their manuscript fully available?

Reviewer #1: Yes

Reviewer #3: Yes

5. Is the manuscript presented in an intelligible fashion and written in standard English?

Reviewer #1: Yes

Reviewer #3: Yes

6. Review Comments to the Author

Reviewer #1: I would like to thank the authors for revising the manuscript. You adequately addressed most of my comments.

1- However, it seems that I didn’t make my first previous comment clear. I was not commenting on mortality of different ARDS severities. I was commenting on independent predictors of mortality in ARDS in general, which is the primary outcome of the study.

So, when you say in the abstract:

Page 5 & 6 (lines 85 – 87): Both oxygenation and driving pressure were independently associated with mortality in ARDS (p=0.014).

You give the impression that each one separately is a predictor of mortality, which is not the case. According to table S5: It is the combined driving pressure and P/F ratio.

So, please make it clear: driving pressure > 14 and P/F ratio ≤ 150 was an independent predictor of mortality (OR = 2.497(95% CI: 1.201-5.191); p = 0.014).

2- Table S5 is the same as table 3 in the manuscript, with the addition of 4 categories of different combinations of driving pressure and P/F ratio, so why not simply add those categories to table 2.

3- Page 17, lines 274-277: You still refer to results of Kaplan Meier curves and Log rank tests as (Mortality rates). It is Survival.

4- In the conclusion (page 23, lines 375, 376) you stated that CCI is an independent predictor of outcome, when it isn’t: OR = 1.072[95% CI: (0.985-1.166)]; p = 0.106

I think that your conclusion should be: Driving pressure alone, or in combination with P/F ratio are independent predictors of mortality.

Reviewer #3: dear respectable authors

thank you for your work and finalizing the recommended modifications.

7. PLOS authors have the option to publish the peer review history of their article (what does this mean?). If published, this will include your full peer review and any attached files.

Reviewer #1: No

Reviewer #3: **Yes: **Mohamed E Abuelnaga, MD

---

## [Author Response · Author response to Decision Letter 1]

14 Nov 2023

Responses to Reviewers (PONE-D-23-11371R1)

Responses to the Reviewers

We would like to express our gratitude to the reviewers for their thorough evaluation of our manuscript and for providing valuable comments and suggestions. We have carefully addressed all of their remarks, as detailed below, and provided our responses to specific comments. The revised sections are highlighted in red font. We believe that, as a result of the reviewers' recommendations, our manuscript has become clearer, and we hope that these changes will be deemed satisfactory.

Reviewer #1: I would like to thank the authors for revising the manuscript. You adequately addressed most of my comments.

Responses to Reviewer #1

Comment 1-1:

1- However, it seems that I didn’t make my first previous comment clear. I was not commenting on mortality of different ARDS severities. I was commenting on independent predictors of mortality in ARDS in general, which is the primary outcome of the study.

So, when you say in the abstract:

Page 5 & 6 (lines 85 – 87): Both oxygenation and driving pressure were independently associated with mortality in ARDS (p=0.014).

You give the impression that each one separately is a predictor of mortality, which is not the case. According to table S5: It is the combined driving pressure and P/F ratio.

So, please make it clear: driving pressure > 14 and P/F ratio ≤ 150 was an independent predictor of mortality (OR = 2.497(95% CI: 1.201-5.191); p = 0.014). 

Response: 

 We appreciate the reviewer for reminding us of this important issue. The statement in abstract section is revised as following: In a multivariate logistic regression model, combination of driving pressure (DP) > 14cmH2O and oxygenation (P/F ratio) ≤150 was an independent predictor of mortality (OR 2.497, 95% CI 1.201-5.191, p=0.014). (Page 5, lines 82 – 85)

Comment 1-2:

2- Table S5 is the same as table 3 in the manuscript, with the addition of 4 categories of different combinations of driving pressure and P/F ratio, so why not simply add those categories to table 2.

Response: 

 We thank the reviewer for this valuable suggestion. We have revised Table 2 and Table 3. According to the reviewer`s suggestion, we also delete Table S5, as it is similar to Table 3, and added 4 categories. We provide a statement to clarify this issue. The manuscript is also revised as following:

1. In the Abstract section: In a multivariate logistic regression model, combination of driving pressure (DP) > 14cmH2O and oxygenation (P/F ratio) ≤150 was an independent predictor of mortality (OR 2.497, 95% CI 1.201-5.191, p=0.014).(page 5, line 82-85)

2. In the Result section:

(1) We further divided the patients into four groups by DP and oxygenation. Patients with DP>14 and P/F ratio≤150 had the highest mortality rate (31/52= 60.8%) (Table 2). (page 14, line 226-228)

(2) A multivariate regression model was adopted to adjusting for possible confounders, including prone position ventilation, aspiration of gastric content, CCI, RR, FiO2, mean arterial pressure, BMI, APACHE II score and DP and P/F ratio combination groups. (page 16, line 258-261)

(3) Two independent factors were identified, including APACHE II score (adjusted odds ratio (OR) 1.089, 95% confidence interval (CI) 1.045-1.135; p<0.0001) and combination of DP/oxygenation (DP>14/P/F ratio≤150, OR 2.497, 95% CI 1.201-5.191, p=0.014) (Table 3). (page 16, line 262-266)

(4) Additionally, our study identified combination of oxygen and respiratory mechanics, specifically in DP>14 and P/F ratio≤150, as an independent risk factor (Table 3, p=0.014). (page 16, line 269-271)

3. In the Discussion section: We found APACHE II scores and a combination of oxygenation and driving pressure were independent mortality risk factors. A combination of oxygenation status and respiratory mechanics gives better outcome prediction in ARDS patients. (page 18, line 282-285)

4. In the Conclusion section: ARDS is common among critically ill patients who are admitted to the ICU and who use invasive mechanical ventilation. Hospital mortality remains high and could potentially be improved by adherence to currently available evidence. Combining oxygenation status and respiratory mechanics may better predict ARDS patients at risk of mortality. (page 23, line 373-377)

5. Comment 1-3:

3- Page 17, lines 274-277: You still refer to results of Kaplan Meier curves and Log rank tests as (Mortality rates). It is Survival.

Response: 

We would like to thank the reviewer for the suggestion to our manuscript. In response, we revised the statement in the Result section as following: Patients with a driving pressure >14 cmH2O (Fig 2A, p=0.025) and a P/F ratio ≤150 (Fig2B, p=0.002) had a lower hospital survival rate. By combining these factors, patients with worse oxygenation and higher driving pressure had the lowest hospital survival rate (Fig 2C, p<0.0001). (page 17, line 274-277)

4- In the conclusion (page 23, lines 375, 376) you stated that CCI is an independent predictor of outcome, when it isn’t: OR = 1.072[95% CI: (0.985-1.166)]; p = 0.106

I think that your conclusion should be: Driving pressure alone, or in combination with P/F ratio are independent predictors of mortality.

Response: 

We thank the reviewer for bringing this detail to our attention. In response to the reviewer’s suggestion, we have revised the statement in the Conclusion section as follows: 

ARDS is common among critically ill patients who are admitted to the ICU and who use invasive mechanical ventilation. Hospital mortality remains high and could potentially be improved by adherence to currently available evidence. Combining oxygenation status and respiratory mechanics may better predict ARDS patients at risk of mortality. (page 23, line 373-377)

Responses to Journal Requirements (PONE-D-23-11371R1)

Journal Requirements:

Response:

 Removed references:

 We remove these references as per the previous reviewer`s suggestion (PONE-D-23-11371) to stay on topic and avoid discussing unrelated issues. Therefore, we have removed all content related to DNR from the article, including the reference citation. 

 22. Phua, J., G.M. Joynt, M. Nishimura, Y. Deng, S.N. Myatra, Y.H. Chan, et al. Withholding and withdrawal of life-sustaining treatments in low-middle-income versus high-income Asian countries and regions. Intensive Care Med, 2016; 42(7): 1118-27.

 23. Lautrette, A., M. Garrouste-Orgeas, P.M. Bertrand, D. Goldgran-Toledano, S. Jamali, V. Laurent, et al. Respective impact of no escalation of treatment, withholding and withdrawal of life-sustaining treatment on ICU patients' prognosis: a multicenter study of the Outcomerea Research Group. Intensive Care Med, 2015; 41(10): 1763-72.

 Added references:

 We included these references for the reviewer`s suggestions to cite within the first 7 lines of the Introduction section of the manuscript.

 1. Thompson, B.T., R.C. Chambers, and K.D. Liu. Acute Respiratory Distress Syndrome.NEnglJMed,2017;377(6):562-572.https://doi.org/10.1056/NEJMra1608077 PMID:28792873.

 2. Ware, L.B. and M.A. Matthay. The acute respiratory distress syndrome. NEnglJMed,2000;342(18):1334-49.https://doi.org/10.1056/nejm200005043421806 PMID:10793167.

 3. Cardinal-Fernández, P., J.A. Lorente, A. Ballén-Barragán, and G. Matute-Bello. Acute Respiratory Distress Syndrome and Diffuse Alveolar Damage. New Insights on a Complex Relationship. AnnAm Thorac Soc, 2017; 14(6):844-850.https://doi.org/10.1513/AnnalsATS.201609-728PS PMID:28570160.

 4. Matthay, M.A., R.L. Zemans, G.A. Zimmerman, Y.M. Arabi, J.R. Beitler, A. Mercat, et al. Acute respiratory distress syndrome. Nat Rev Dis Primers, 2019; 5(1): 18. https://doi.org/10.1038/s41572-019-0069-0 PMID:30872586.

 5. Sweeney, R.M. and D.F. McAuley. Acute respiratory distress syndrome. Lancet,2016;388(10058):2416-2430.https://doi.org/10.1016/s0140-6736(16)00578-x PMID:27133972.

---

## [Decision Letter · Decision Letter 2]

20 Nov 2023

A combination of oxygenation and driving pressure can provide valuable information in predicting the risk of mortality in ARDS patients.

PONE-D-23-11371R2

Dear Dr. Chan,

We’re pleased to inform you that your manuscript has been judged scientifically suitable for publication and will be formally accepted for publication once it meets all outstanding technical requirements.

Kind regards,

Academic Editor

PLOS ONE

Additional Editor Comments (optional):

Reviewers' comments:

Reviewer's Responses to Questions

**Comments to the Author**

1. If the authors have adequately addressed your comments raised in a previous round of review and you feel that this manuscript is now acceptable for publication, you may indicate that here to bypass the “Comments to the Author” section, enter your conflict of interest statement in the “Confidential to Editor” section, and submit your "Accept" recommendation.

Reviewer #1: All comments have been addressed

Reviewer #3: All comments have been addressed

2. Is the manuscript technically sound, and do the data support the conclusions?

Reviewer #1: Yes

Reviewer #3: Yes

3. Has the statistical analysis been performed appropriately and rigorously? 

Reviewer #1: Yes

Reviewer #3: Yes

4. Have the authors made all data underlying the findings in their manuscript fully available?

Reviewer #1: Yes

Reviewer #3: Yes

5. Is the manuscript presented in an intelligible fashion and written in standard English?

Reviewer #1: Yes

Reviewer #3: Yes

6. Review Comments to the Author

Reviewer #1: Thank you for revising the manuscript, and addressing my concerns.

The article has substantially improved.

Congratulations.

Reviewer #3: dear respectable authors

thank you for your valuable work

7. PLOS authors have the option to publish the peer review history of their article (what does this mean?). If published, this will include your full peer review and any attached files.

Reviewer #1: No

Reviewer #3: **Yes: **Mohamed E Abuelnaga, MD

---

## [Editor Report · Acceptance letter]

1 Dec 2023

PONE-D-23-11371R2 

A combination of oxygenation and driving pressure can provide valuable information in predicting the risk of mortality in ARDS patients. 

Dear Dr. Chan:

I'm pleased to inform you that your manuscript has been deemed suitable for publication in PLOS ONE. Congratulations! Your manuscript is now with our production department. 

Kind regards, 

on behalf of

Dr. Robert Jeenchen Chen 

Academic Editor

PLOS ONE